# Endophytic Fungi in Rice Plants and Their Prospective Uses

Yingping Hu [1] , Guodong Lu [2,3], Dongmei Lin [3,4], Hailin Luo [3,4], Mediatrice Hatungimana [1,5], Bin Liu [4,6,*] and Zhanxi Lin [3,4,*]

1   College of Life Sciences, Fujian Agriculture and Forestry University, Fuzhou 350002, China;
    hyp111419@126.com (Y.H.); mediatunga@gmail.com (M.H.)
2   College of Plant Protection, Fujian Agriculture and Forestry University, Fuzhou 350002, China;
    gdlufafu@163.com
3   National Engineering Research Center of Juncao Technology, Fuzhou 350002, China;
    lindongmei@fafu.edu.cn (D.L.); lhljuncao@fafu.edu.cn (H.L.)
4   College of Juncao Science and Ecology, Fujian Agriculture and Forestry University, Fuzhou 350002, China
5   Rwanda Agriculture and Animal Resources Development Board, Huye P.O. Box 5016, Rwanda
6   College of Food Science, Fujian Agriculture and Forestry University, Fuzhou 350002, China
*   Correspondence: 000q080003@fafu.edu.cn (B.L.); 000q120034@fafu.edu.cn (Z.L.); Tel.: +86-13328661151 (B.L.);
    +86-13705039961 (Z.L.)

**Abstract:** In a long-term evolution, fungal endophytes have formed a mutually beneficial relationship with host plants. Therefore, what roles do fungal endophytes play in the growth and development of rice, one of the major food crops in the world, and agricultural production? This mini-review aims to highlight the diversity, identification, colonization, function, and mechanism of action of endophytic fungi isolated from rice tissues through a literature review; comprehensively expound the interaction mechanism between rice fungal endophytes and their hosts in stimulating the growth of rice plants and alleviating biological and abiotic stresses on plants; and contribute new ideas for rice production and a sustainable rice industry. Additionally, rice fungal endophytes, as a new resource, have broad prospects in the development of biopesticides, biocontrol agents, and new medicine.

**Keywords:** rice; endophytic fungi; diversity; function; secondary metabolites

## 1. Introduction

Micro-organisms have a long-standing interaction with plants throughout the plant life cycle. Some have adverse effects on the plant host, whereas others are not harmful or are beneficial to the host, e.g., stimulating plant host growth [1,2]. In a long-term evolution, there is a special type of micro-organism in the plant tissue called the endophyte, which exists or colonizes among plant tissues but will not harm or adversely affect the host plants, forming a mutually beneficial symbiotic relationship with the host [3]. The endophytes comprise both endophytic bacteria and endophytic fungi. Many studies have demonstrated that plant endophytes play a variety of beneficial roles in the host, including the production of plant hormones or metabolites to stimulate plant growth, promotion of nutrient absorption and utilization, biological nitrogen fixation, stress reduction, resistance enhancement, and induction of the plant's natural immune system to protect the plant host [4–9]. As a type of eukaryotic micro-organism, endophytic fungi, which include endophytic filamentous fungi and endophytic yeast, naturally settle in the internal tissues of healthy plants and will not cause any harm to the host [10]. Fungal endophytes exist in almost all healthy plant tissues on earth with high community diversity [9,11], establishing symbiotic, commensal, and parasitic relationships with host plants through long-term evolution. Endophytic fungi acquire nutrition from plants. In return, they can provide benefits for the survival of host plants by synthesizing certain metabolites, stimulating the growth and development of host plants, and helping plants tolerate biotic and abiotic stresses by regulating their immunity and stimulating the production of

metabolites [12–18]. In addition, fungal endophytes can secrete some metabolic substances that antagonize bacteria, providing positive guidance for agricultural production. Depsidone compounds isolated from endophytic fungi exhibit good antibacterial effects against Gram-positive bacterial strains, such as methicillin-resistant *Staphylococcus aureus* [MRSA], *Bacillus subtilis*, and *S. aureus* [19]. Sesquiterpene compounds obtained from endophytic fungus *Colletotrichum* sp. B-89 have antibacterial activities against *S. aureus*, *B. subtilis*, and *Lebsiella pneumoniae* [20]. The *Fusarium* species are ubiquitous in herbaceous plants. *Fusarium fujikuroi* complex species are a group species of the genus *Fusarium*, which are associated with Bakanae disease in rice, and they are becoming a serious concern for the good yield and quality of rice [21,22]. But some fungal endophytes of the *Fusarium* complex play an important role in enhancing the host's adaptability to biotic and abiotic stress [23]. As one of the major food crops in the world, rice feeds more than half of the world's population [24]. With the increasing world population, global climate change, the abuse of pesticides and fertilizers causing environmental pollution, hampering rice productivity, and many other challenges, the human living environment is encountering unprecedented challenges, and there is a surge in demand for healthy foods [25–29]. To realize the sustainable development of agriculture and maintain the ecological balance, it is crucial to lessen the excessive use of chemical fertilizers and pesticides and ensure grain output. People are attempting to lessen the excessive use of chemical fertilizers and improve the disease resistance of crops through some new biological fertilizers, biological agents, and measures to improve the yield and quality of crops [8,30]. As of now, the scientific community is searching for a greener, more efficient, and sustainable substitute for enhancing plants' resistance. Endophytic fungi are being developed and studied as a biological resource to effectively alleviate the damage of environmental stress from plants. In order to gain a clearer understanding of the research on endophytic fungi in rice, this paper mainly reviewed the research status of rice endophytic fungi and focused on the diversity, colonization, function, and mechanism of action between the host and endophytic fungi for converting the traditional rice cultivation mode to a sustainable development mode.

## 2. Isolation and Identification of Endophytic Fungi in Rice

Culture-dependent and culture-independent methods are applied to study endophytic fungi diversity in rice tissues. Plenty of rice plant endophytic fungi have been isolated and identified using culture-dependent methods, which are the most common approaches for isolating and culturing endophytic fungi from rice tissues prior to their identification [31–36]. However, in order to study endophytic fungal diversity through these methods, the fungi must be culturable based on different media. But fungal diversity isolated from rice plant samples is fully dependent on the isolation protocol, which can be visualized in Figure 1. Usually, healthy and undamaged samples are collected, kept fresh during transportation, and brought to the laboratory immediately for isolation. The key step is to sterilize the surface of the samples. Typically, samples are selected and rinsed with tap water in a short time. Surface sterilization of rice samples is first conducted by soaking the samples in 70% ethanol, immersing sodium hypochlorite (NaClO), and then washing the samples in 70% ethanol solution again, using sterile distilled water to remove traces of chemical solutions from samples at the end. But the concentration of NaClO and the disinfection time used for surface disinfection vary for different samples in different studies. Low-concentration NaClO has a relatively longer disinfection time when used in rice tissues [34,35,37]. The tissues are cut into appropriate segments, or the outer cover is removed to be grown in a medium or mixed with aseptic tetracycline (50 mg/L) for incubation for 7~10 days at room temperature; then, the endophytic fungal colonies are sub-cultured in fresh medium to obtain pure cultures [31,32,35–39]. The purified isolates are transferred to potato dextrose agar (PDA) or yeast extract–malt extract (YM) slants for culturing and stored at 4 °C [31].

The culturable fungi identification of rice endophytic fungi is classified into operational taxonomic units (OTUs) according to the culture characteristics and morphological, biochemical, and genetic characteristics. Normally, an optical microscope and a low-temperature scanning electron microscope are used to observe the microscopic and morphological characteristics (colony appearance, growth rate, conidial morphology, and conidia/spore structure) of the strains after incubation [31,40].

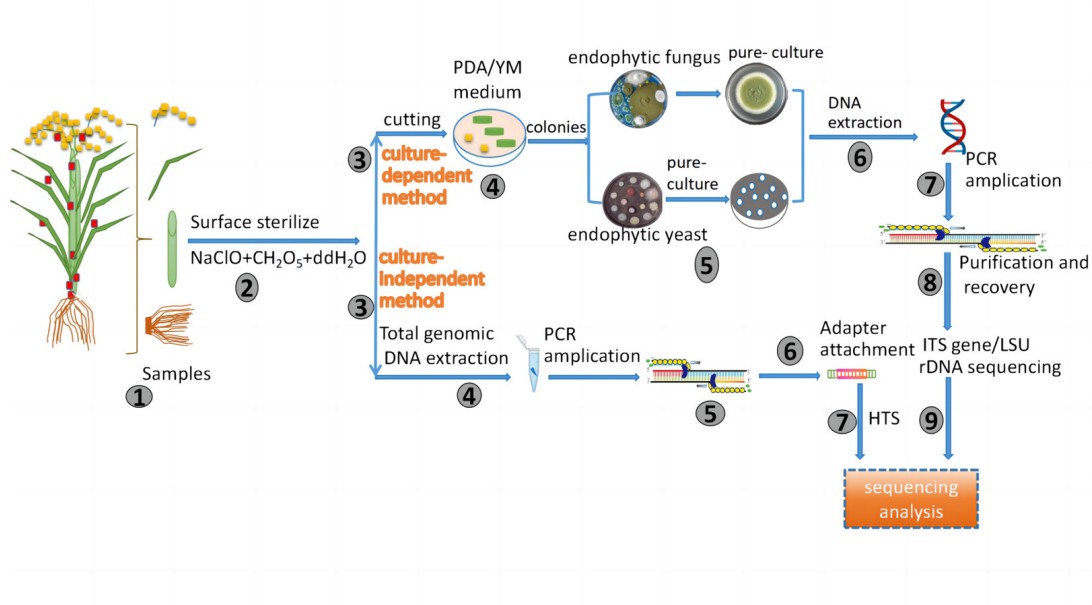

**Figure 1.** Methods of isolation and identification of rice fungal endophytes.

Due to unknown culture conditions, including environmental factors and nutrients, some endophytic fungi cannot be isolated from plants by culture-dependent methods [41]. Culture-independent methods are available for the exploration of fungal communities in rice plants. The key step in the culture-independent strategy—which relies on genomic DNA (gDNA) extraction from surface-sterilized plant tissues and polymerase chain reaction (PCR) amplification of selected sequences—is used to identify the endophytic fungal communities (see the procedure outlined in Figure 1). Moreover, to avoid bacterial 16 S rRNA gene contamination, the surface sterilization protocol with formaldehyde (36%) and 0.1 M sodium hydroxide (NaOH) solution can be strongly effective in removing all rhizoplane bacteria [42].

Molecular techniques are necessary and reliable in identifying fungal endophytes. For filamentous fungal endophytes, the internal transcribed spacer (ITS) sequencing method is currently the most common molecular identification method for endophytic fungi taxonomic classification. The ITS region (ITS1–5.8S–ITS2) is amplified by PCR from genomic DNA using the ITS1 primer (5′-CTTGGTCATTTAGAGGAAGTAA-3′) and ITS4 primer (5′-TCCTTATTGATATGC-3′) [31,35,43,44]. But for endophytic yeast, primers NL1 and NL4 are used to amplify the gene encoding the D1/D2 region of the large subunit (LSU) rRNA from genomic DNA [34,45]. The amplified products are quantified and sequenced, and the sequencing results are searched for homology using the National Center for Biotechnology Information (NCBI) BLAST program (http://blast.ncbi.nlm.nih.gov (accessed on 25 April 2024)). The identification of endophytes is in accordance with the maximum query coverage and score in the BLAST results [31,46], and the phylogenetic analysis uses bioinformatics tools [35].

### 3. Variations in Endophytic Fungal Communities in Rice: Exploring the Diversity and Factors Influencing Their Composition

Endophytes exist in almost all plants on earth, and fungi are abundant in every host plant examined [12,47]. Rice consists of a high diversity of endophytic fungi in most parts of the plant. The types and numbers of endophytic fungi in different parts of rice plants are various [36]. Wang et al. encountered the highest diversity of fungal OTUs in rice plant buds, followed by stems, and the lowest diversity of endophytic fungi communities was found in roots [42]. Also, the highest number of endophytic fungi was detected in rice seeds, followed by leaves, roots, and stems [38]. Seephueak et al. found that the diversity of endophytic fungi was highest during the peak tillering stage of rice plant's growth and development [48]. Other studies have shown that the number of endophytic fungi in the roots of first-generation rice plants acquired by crossing wild rice with cultivated rice was more than that of their parents [49].

A total of 494 endophytic fungal isolates that were isolated from 720 samples of leaves, roots, and seeds from salt-sensitive and salt-tolerant rice varieties—with the highest number of isolates from seeds—were categorized into 41 OTUs [31]. Endophytic fungi were found in rice roots, stems, leaves, and grains through a culture-dependent method, mainly concentrated in more than 60 genera. *Penicillium*, *Aspergillus*, and *Fusarium* are the most common fungi in rice plants [10,31–33,38,40,43,44,48,50–57], which can be visualized in Figure 2. Herein, we conducted statistical analysis on the percent of identified isolates through a culture-dependent method from the literature and found that they are mainly distributed in five phyla using the Origin 2021 software, namely *Ascomycota*, *Basidiomycota*, *Mucoromycota*, *Oomycota*, and *Zoopagomycota*, with *Ascomycota* being the main group (Figure 3). Data were entered into SPSS ver. 26 and analyzed using single-factor analysis of variance (ANOVA) and multiple comparisons (LSD) tests, and the results showed that the mean percent of *Basidiobolus* in the leaf is significantly higher than that in the root and seed. To explore endophytic fungal communities in the root during the growth of Jiafuzhan ratooning rice, 497 amplicon sequence variants (ASVs) were obtained by sequencing the internal transcribed spacer (ITS) genes, and 12 phyla endophytic fungi were detected. *Ascomycota*, *Basidiomycota*, *Mortierellomycota*, *Rozellpmycota*, and *Mucoromycota* were the most abundant fungal phyla [58]. Diverse fungi and unidentified fungi were detected in rice sprouts by ITS1 and ITS2, and *Ascomycota* and *Basidiomycota* were the dominant fungal phyla [59]. Thirty endophytic fungi were isolated from the roots of upland rice, and *Talaromyces pinophilus* was the most dominant species in the fungal community [60]. In addition, culturable endophytic yeast isolates were distributed among 19 species, including 12 species in the nine genera of *Basidiomycota* and 7 species in the five genera of *Saccharomycota*. *Pseudozyma churashimaensis* was the most abundant species in the yeast community of rice leaves [34]. On a culture-independent basis, the diversity of the endophytic yeasts in rice leaves revealed that the majority of the yeasts were basidiomycetous yeasts [41].

Endophytic fungal diversity is affected by many factors, such as environmental factors, cultivation methods, nutrient availability, host plant genotype, and so on [40,49,59,61,62]. The abundance of fungal phyla changes throughout the growth stages of rice [58]. Co-occurrence network analysis showed that the bacterial community exerted a positive influence on fungal endophytes in rice plants [43]. *Gibberella intermedia*, *Fusarium solani*, and a few endophytic fungi, such as *Botryosphaeria dothidea* and *Alternaria alternata*, can only be isolated from the salt-tolerant rice variety Pokkali [31]. Due to the limitations of certain methods, some strains are difficult to detect. Some species cannot be found using culture-independent approaches in some research works but are identified using culture-dependent approaches in other research works. Therefore, a combination of culture-dependent and culture-independent approaches might result in more complete information regarding the fungal endophyte communities in the tissues of rice plants [34].

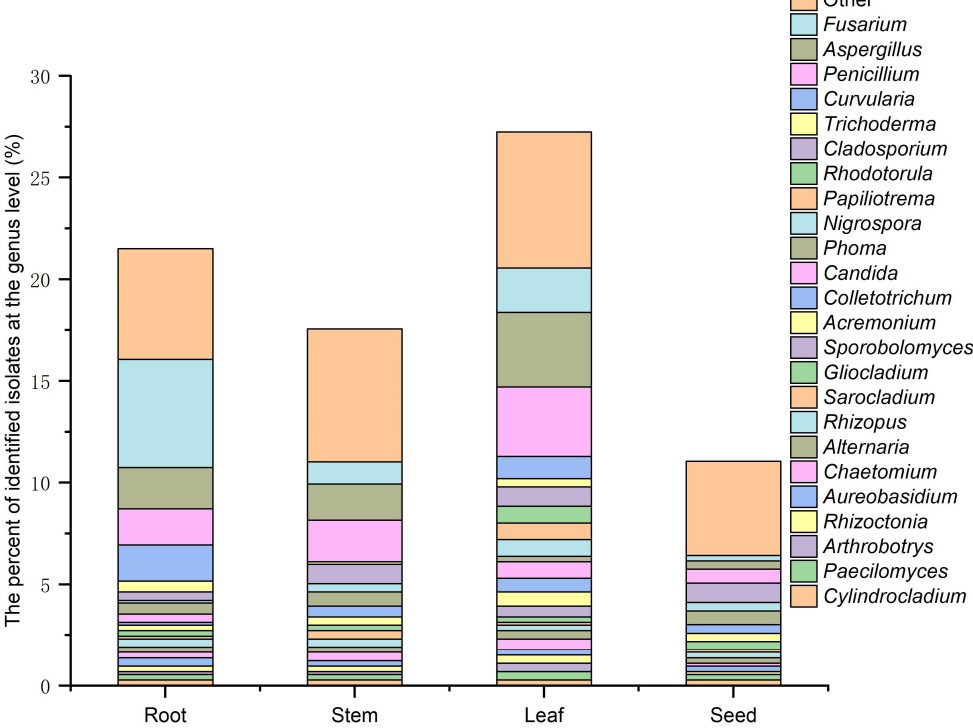

**Figure 2.** The percent of 25 kinds of endophytic fungi in different tissues of rice at the genus level.

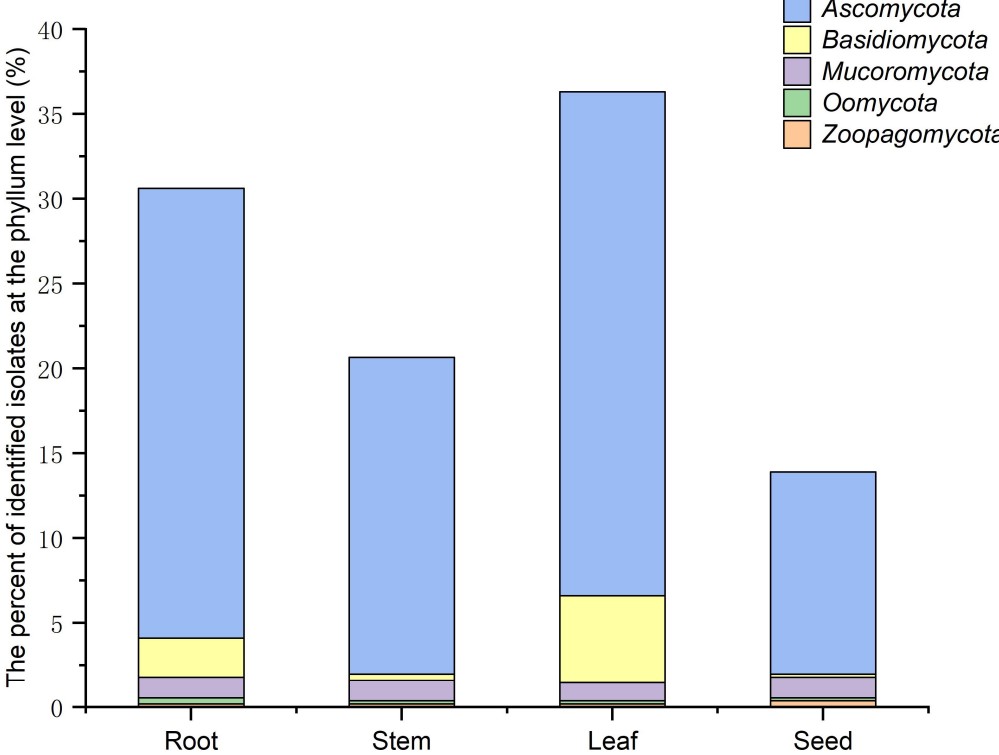

**Figure 3.** The percent of endophytic fungi in different tissues of rice at the phylum level.

Noteworthily, bioinformatics analysis plays a crucial role in fungal diversity analysis. High-throughput amplicon sequencing (HTAS) and bioinformatics tools have been well applied in analyzing fungal diversity and interpreting the results perfectly, breaking through the shortcomings of traditional culture-dependent methods, which are difficult in application when culturing some micro-organisms [63–65]. The Divisive Amplicon Denoising Algorithm (DADA2) and the Quantitative Insights Into Microbial Ecology (QIIME) are helpful for data processing and data analysis [66,67]. Based on clustering the sequences into amplicon variant sequences (ASVs), the taxonomy can be assigned to the ASVs [58,65,68]. Typically, alpha diversity and beta diversity analyses are performed using the diversity plugin. The alpha diversity is used to analyze the description of the endophytic fungal community. The beta diversity analysis is performed to investigate the structural variation in fungal communities among the samples. Bioinformatics analysis also plays an important role in characterizing the biosynthetic potential of secondary metabolites in endophytic fungi [69]. These tools are practically well applied to endophytic fungal communities of rice plants [34,42,58,59].

## 4. Colonization of Endophytic Fungi in Rice

Endophytic fungi can produce a type of cell wall degrading enzymes, such as cellulase, laccase, pectinase, and xylanase, through metabolism to degrade plant cell walls and change their structure, thereby colonizing and thriving in plant tissues [59,70]. Spore suspension and plate methods can be used for the colonization of fungal endophytes in rice plants [32]. Some studies show that endophytic fungi can colonize the roots, buds, stems, leaves, sheathes, grains, and other parts of rice plants [32,42,54,60,71–73]. Dark septate endophytes (DSEs) can colonize the root cortex of rice, forming structures resembling anastomoses and intracellular microsclerotia [71,74]. An inoculation experiment indicated that endophytic fungus *Phomopsis liquidambari* B3 was inoculated with a green fluorescent protein tag to 10-day-old rice seedlings, and it was found that the strain colonized the roots of rice plants [75]. Aerobic soil conditions may stimulate the colonization of rice roots by endophytic fungal strains [76]. The colonization mode of endophytic fungi in rice plant tissues is similar to that of plant pathogens and mycorrhizal fungi, including fungal host recognition, spore germination, epidermal infiltration, and tissue colonization [32,77–79]. Plants selectively allow fungal colonization through phenotypic genes and metabolic signals to shape the structure of endophytic flora. Accordingly, these fungi generate various adaptations through symbiotic signal transduction, avoid the attack of plant immune signals, and grow as endophytes in plant organs [12,80,81]. The colonization and distribution of fungal endophytes in plant tissues are affected by environmental factors, the interaction between micro-organisms and plants, and the interaction among endophytic communities [82,83]. Phytohormones play a crucial role in the colonization of endophytic fungi in host plants. The colonization of *Acremonium* sp. D212 in the roots of rice was regulated by the concentration of methyl jasmonate (MeJA) and 1-naphthaleneacetic acid (NAA) [78].

## 5. Effects of Endophytic Fungi in Rice Plant

The effects of endophytes are complicated and indirectly or directly produce beneficial effects for their host plant [4,84]. Herein, we discuss and summarize the mechanisms of endophytic fungi in rice plants (Figure 4).

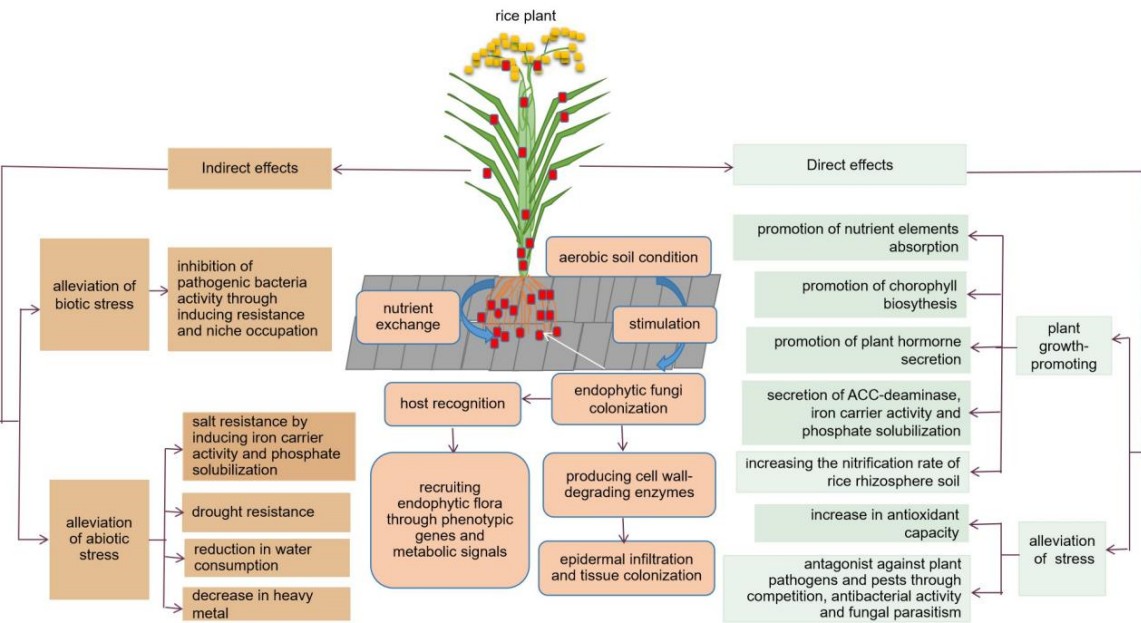

**Figure 4.** Mechanisms of fungal endophytes in rice plant.

*5.1. Direct Effects*

5.1.1. Mineral Uptake

Fungal endophytes can establish reciprocal relationships with rice (*Oryza sativa*) and stimulate the growth of rice plants. The combination of *Acremonium* and *Arthrobotrys* can improve biomass and rice yield under field conditions [32]. The fungal endophyte *Diaporthe liquidambaris* can promote chlorophyll biosynthesis and water-soluble carbohydrate accumulation in rice roots under low N conditions [85]. Endophytic fungi *Trichoderma zelobreve* PBMP16 had a positive effect on upland rice growth and increased the phenolic compounds, anthocyanin, antioxidants, and nutrient N uptake of rice plant compared with those of the non-inoculated control [86,87]. The DSE isolates had the potential for the promotion of rice plant growth to increase the tillering, nutrient N and P uptake, accumulation of nutrients, and growth [71,74]. It is well known that most of the phosphorus in the soil cannot be directly absorbed and utilized by plants, and the fungus associated with the plants can dissolve the mineral P by producing organic acids and help the host plant absorb P [88]. Endophytic fungal isolates from rice plants showed high phosphate solubility with a concentration of 3.719 mg·L$^{-1}$ [36].

It has been shown that some endophytic fungi, such as *P. liquidambari*, can increase the nitrification rate of rice rhizosphere soil, promote the uptake and metabolism of nitrogen by rice plants, and improve the mineral nutrition, quality, and yield of rice in symbiosis with rice plants under the conditions of nitrogen and phosphorus deficiency. Furthermore, some endophytic fungi can not only significantly improve the yield of rice but also significantly increase the accumulation of nitrogen, phosphorus, iron, manganese, zinc, molybdenum, selenium, and protein content in rice seeds [89–92]. These beneficial effects are attributed to the symbiosis with *P. Liquidambari*, which stimulates the upregulation of the genes involved in N uptake and metabolism, promotes the formation of iron plaque, changes the endophytic bacteria community, and increases the concentration of organic compounds (soluble sugar, total free amino acids, and organic acids) in rice root exudates [93–95]. Next-generation sequencing technology was used to compare the gene expression differences between rice seedlings inoculated with *Trichoderma asperellum* SL2 and rice seedlings not inoculated with *T. asperellum* SL2. It was shown that specific genes associated with photosynthesis, RNA activity, stomatal activity, and root development were found to be upregulated in rice seedlings inoculated with *T. asperellum* SL2 [96].

### 5.1.2. Promotion of Plant Hormone Secretion

The symbiotic relationship between endophytic fungi and their hosts involves many metabolic and regulatory processes, and plant hormones play a key role in the establishment of these symbiotic relationships [97]. Endophytic fungi have been tested as plant-growth-promoting fungi (PGPF), which have the ability to produce the Indole-3-acetic acid (IAA) hormone, increasing the growth of rice seedlings [36]. They can be used as plant growth promoters to stimulate rice growth by secreting a large number of plant hormones, including auxin, jasmonic acid, and gibberellin, and inducing iron carrier activity and phosphate solubilization, which can alleviate abiotic stresses, such as salt and drought, and enhance disease resistance [98]. Endophytic fungi like *Acremonium*, *Arthrobotrys*, *Paecilomyces formosus* LWL1, *Phoma glomerata* LWL2, *Penicillium* sp. LWL3, and *Phialemonium dimorphosporum* have the potential to secrete hormones such as gibberellin and IAA, which significantly stimulate the growth of the gibberellin-deficient dwarf mutant Waito-C rice, and can alleviate abiotic stresses, such as salt and drought [32,79,99,100]. *Talaromyces adpressus* (OPCRE2), *Talaromyces argentinensis*, and *Aspergillus welwitschiae* Ocstreb1 (AwOcstreb1) exhibit various plant-growth-promoting (PGP) abilities in IAA, 1-aminocyclop ropane-1-carboxylate deaminase (ACC-deaminase), siderophore production, phosphate, and zinc solubilization; all of these substances are beneficial for rice growth [101]. Rice plants inoculated with *Absidia* and *Cylindrocladium* endophytic fungal isolates showed significant increases ($p \leq 0.05$) in plant height, dry weight, and fresh weight [37]. A total of 1035 endophytic yeast isolates were isolated from rice and sugarcane, and their IAA production was determined. The results showed that 167 isolates were IAA producers [45]. Many endophytic fungi isolated from rice and upland rice can promote the production of indoleacetic acid substances to stimulate plant growth and increase the stem length, fresh weight of stem and root, antioxidant capacity, proline, and soluble sugar content of rice plants [60,102,103] (Table 1).

**Table 1.** Endophytic fungi isolated from various parts of rice plants.

| Rice Variety | Phytonic Habitat | Endophytic Fungal Taxa | Reference |
|---|---|---|---|
| Wild rice | Roots | *Falciphora oryzae* (a DSE strain) [i] | [18] |
| Traditional rice variety in Sri Lanka | Seeds, stems, leaves | *Acremonium* | [32] |
| Traditional rice variety in Sri Lanka | Roots, stems, leaves | *Arthrobotrys* | [32] |
| Traditional rice variety in Sri Lanka | Roots | *Aspergillus* | [32] |
| Traditional rice variety in Sri Lanka | Roots | *Aureobasidium* | [32] |
| Traditional rice variety in Sri Lanka | Roots | *Chaetomium* | [32] |
| Traditional rice variety in Sri Lanka | Stems, leaves, seeds | *Colletotrichum* | [32] |
| Traditional rice variety in Sri Lanka | Roots | *Curvularia* | [32] |
| Traditional rice variety in Sri Lanka | Seeds | *Fusarium* | [32] |
| Traditional rice variety in Sri Lanka | Roots, stems, leaves | *Humicola* | [32] |
| Traditional rice variety in Sri Lanka | Roots, stems | *Penicillium* | [32] |
| Traditional rice variety in Sri Lanka | Roots, stems | *Phoma* | [32] |
| Traditional rice variety in Sri Lanka | Leaves | *Rhizoctonia* | [32] |

**Table 1.** *Cont.*

| Rice Variety | Phytonic Habitat | Endophytic Fungal Taxa | Reference |
|---|---|---|---|
| Traditional rice variety in Sri Lanka | Stems | *Rhizopus* | [32] |
| Traditional rice variety in Sri Lanka | Roots | *Trichoderma* | [32] |
| Paddy rice (Oryza sativa) | Seeds | *Fusarium, Aspergillus, Curvularia* | [33] |
| Paddy rice (Oryza sativa) | Roots | *Fusarium, Penicillium, Gilmaniella* | [33] |
| Paddy rice (Oryza sativa) | Leaves | *Fusarium, Aspergillus, Curvularia, Penicillium, Arthrobotrys foliicola* | [33] |
| Paddy rice (Oryza sativa) | Stems | *Fusarium, Aspergillus, Penicillium* | [33] |
| *O. sativa* L. | Leaves | *Saitozyma flava* | [34] |
| *O. sativa* L. | Leaves | *Papiliotrema japonica* | [34] |
| *O. sativa* L. | Leaves | *Papiliotrema siamense* | [34] |
| *O. sativa* L. | Leaves | *Rhodotorula taiwanensis* | [34] |
| *O. sativa* L. | Leaves | *Rhodotorula* aff. *toruloides* | [34] |
| *O. sativa* L. | Leaves | *Sporobolomyces carnicolor* | [34] |
| *O. sativa* L. | Leaves | *Cystobasidium* aff. *slooffiae* | [34] |
| *O. sativa* L. | Leaves | *Moesziomyces antarcticus* | [34] |
| *O. sativa* L. | Leaves | *Pseudozyma churashimaensis* | [34] |
| *O. sativa* L. | Leaves | *Candida metapsilosis* | [34] |
| *O. sativa* L. | Leaves | *Candida tropicalis* | [34] |
| *O. sativa* L. | Leaves | *Meyerozyma caribbica* | [34] |
| *O. sativa* L. | Leaves | *Wickerhamomyces anomalus* | [34] |
| *O. sativa* L. | Leaves | *Candida citri* | [34] |
| *O. sativa* L. | Leaves | *Kodamaea ohmeri* | [34] |
| *O. sativa* L. | Leaves | *Diutina siamensis* | [34] |
| *O. sativa* L. | Leaves | *Paecilomyces tenuis* EF1 * | [35] |
| *O. sativa* L. | Leaves | *Talaromyces pinophilus* EF2 * | [35] |
| *O. sativa* L. | Leaves | *N. sphaerica* EF3 * | [35] |
| *O. sativa* L. | Leaves | *N. oryzae* EF4 * | [35] |
| *O. sativa* L. | Leaves | *Trichoderma longibrachiatum* EF5 * | [35] |
| *O. sativa* L. | Leaves | *Aspergillus terreus* EF6 * | [35] |
| *O. sativa* L. | Leaves | *T. longibrachiatum* EF7 * | [35] |
| Suwandel rice variety | Leaves, stems, roots, seeds | *Absidia* *, *Cylindrocladium* *, *Aspergillus* *, *Penicillium* *, *Paecilomyces* *, *Aureobasidium*, *Rhizoctonia* *, *Mortierella*, *Fusarium*, *Gliocladium* *, *Phoma* *, *Acremonium* *, *Arthroderma*, *Varicosporium*, *Cladosporium*, *Rhziophus*, *Emericella* | [37] |
| Kaluheenati rice variety | Leaves, stems, roots, seeds | *Colletotrichum* SM, *Absidia* *, *Cylindrocladium* *, *Aspergillus* *, *Penicillium* *, *Paecilomyces* *, *Aureobasidium*, *Rhizoctonia* *, *Mortierella*, *Fusarium*, *Gliocladium* *, *Phoma* *, *Acremonium* *, *Arthroderma*, *Varicosporium*, *Cladosporium*, *Rhziophus* | [37] |
| Wild rice (*O. rufipogon* Griff.) | Seeds | *Cladosporium* sp. * | [38] |
| Wild rice (*O. rufipogon* Griff.) | Seeds | *Alternaria* sp. * | [38] |
| Wild rice (*O. rufipogon* Griff.) | Seeds | *Dendryphiella* sp. * | [38] |
| Wild rice (*O. rufipogon* Griff.) | Seeds | *Phoma* sp. *, *Leptosphaerulina* sp., *Dendryphiella* sp., *Pleosporales* sp., *Phoma* sp. | [38] |
| Wild rice (*O. rufipogon* Griff.) | Stems | *Penicillium* sp. * | [38] |

**Table 1.** *Cont.*

| Rice Variety | Phytonic Habitat | Endophytic Fungal Taxa | Reference |
|---|---|---|---|
| Wild rice (*O. rufipogon* Griff.) | Roots | *Trichoderma* sp. *, *Monographella*, *Bionectria* sp. | [38] |
| Wild rice (*O. rufipogon* Griff.) | Stems | *Sarocladium* sp. * | [38] |
| Wild rice (*O. rufipogon* Griff.) | Stems | *Fusarium* *, *Penicillium* sp., *Gaeumannomyces* | [38] |
| Wild rice (*O. rufipogon* Griff.) | Leaves | *Bipolaris* sp. *, *Paraphaeosphaeria* sp., *Acrophialophora* | [38] |
| Rice variety Ld 368 | Healthy plant parts | *Penicillium* | [39] |
| Rice variety Ld 368 | Healthy plant parts | *Aspergillus* | [39] |
| Rice variety Ld 368 | Healthy plant parts | *Fusarium* sp.* | [39] |
| Rice variety Ld 368 | Healthy plant parts | *Colletotrichum* | [39] |
| Rice variety Ld 368 | Healthy plant parts | *Curvularia* | [39] |
| Rice variety Ld 368 | Healthy plant parts | *Chaetomium* sp.* | [39] |
| Rice variety Ld 368 | Healthy plant parts | *Trichoderma* sp.* | [39] |
| Rice 'Tianyou Huazhan' | Roots | *Pleosporales* sp., *Aspergillus* sp., *Penicillium* sp., *Chaetomium Sordariales*(*o*), *Cladosporium*, and *Apodus* | [43] |
| *O. sativa* L. | Stems, leaves, grains | *C. globosum* | [48] |
| *O. sativa* L. | Stems, leaves | *Acremonium hansfordii* | [48] |
| *O. sativa* L. | Stems | *Acremonium luzulae* | [48] |
| *O. sativa* L. | Stems | *Alternaria alternata* | [48] |
| *O. sativa* L. | Stems, leaves, grains | *A. padwickii* | [48] |
| *O. sativa* L. | Stems, leaves | *A. flavus* | [48] |
| *O. sativa* L. | Stems, leaves | *Aspergillus fumigatus* | [48] |
| *O. sativa* L. | Stems, leaves | *A. niger* | [48] |
| *O. sativa* L. | Stems, leaves, grains | *Bipolaris oryzae* | [48] |
| *O. sativa* L. | Stems, leaves, grains | *Cercospora oryzae* | [48] |
| *O. sativa* L. | Stems, leaves | *Cladosporium cladosporioides* | [48] |
| *O. sativa* L. | Stems, leaves, grains | *Cladosporium elatum* | [48] |
| *O. sativa* L. | Stems, leaves, grains | *Cladosporium oxysporum* | [48] |
| *O. sativa* L. | Stems, leaves, grains | *Cladosporium tenuissimum* | [48] |
| *O. sativa* L. | Stems, leaves, grains | *Colletotrichum graminicola* | [48] |
| *O. sativa* L. | Stems, leaves, grains | *C. lunata* | [48] |
| *O. sativa* L. | Stems, leaves, grains | *Dactylaria hawaiiensis* | [48] |
| *O. sativa* L. | Stems, leaves, grains | *Drechslera australiensis* | [48] |
| *O. sativa* L. | Stems, leaves, grains | *F. solani* | [48] |
| *O. sativa* L. | Stems, leaves, grains | *F. semitectum* | [48] |
| *O. sativa* L. | Stems, leaves, grains | *Nigrospora oryzae* | [48] |
| *O. sativa* L. | Stems, leaves, grains | *Nigrospora sphaerica* | [48] |
| *O. sativa* L. | Stems | *Penicillium* sp. | [48] |
| *O. sativa* L. | Stems | *Penicillium funiculosum* | [48] |
| *O. sativa* L. | Stems, leaves, grains | *Pyricularia oryzae* | [48] |
| *O. sativa* L. | Stems, leaves | *Rhinocladiella similis* | [48] |
| *O. sativa* L. | Stems, leaves | *Rhynchosporium oryzae* | [48] |
| *O. sativa* L. | Stems | *Sarocladium oryzae* | [48] |
| *O. sativa* L. | Stems | *Stachybotrys dichroa* | [48] |
| *O. sativa* L. | Stems | *Veronaea apiculata* | [48] |
| *O. sativa* L. | Stems, leaves | *Veronaea coprophila* | [48] |
| *O. sativa* L. | Leaves, stems, roots | *Microsphaeropsis arundinis* S59 [#] | [50] |
| *O. sativa* L. | Leaves, stems, roots | *Penicillium rubens* L138 [#] | [50] |

**Table 1.** *Cont.*

| Rice Variety | Phytonic Habitat | Endophytic Fungal Taxa | Reference |
|---|---|---|---|
| *O. sativa* L. | Leaves, stems, roots | *A. flavus* L55 ## | [50] |
| *O. sativa* L. | Leaves, stems, roots | *Eupenicillium javanicum* R57 ## | [50] |
| Indian indigenous rice varieties | Seeds | *Aspergillus* | [51] |
| Indian indigenous rice varieties | Seeds | *Fusarium* *,& | [51] |
| Indian indigenous rice varieties | Seeds | *Gliocladium* | [51] |
| Indian indigenous rice varieties | Seeds | *Penicillium* | [51] |
| Indian indigenous rice varieties | Seeds | *Bipolaris* | [51] |
| Indian indigenous rice varieties | Seeds | *Basidiobolus* | [51] |
| Indian indigenous rice varieties | Seeds | *Mycelia sterila* | [51] |
| (*O.sativa* L.) | Seeds, seedlings | *Phialemonium curvatum* *, *Phaeosphaeriopsis musae* *, *Sarocladium oryzae* sp. *, *Penicillium citrinum* *, *Sordariomycetes* sp. *, *Penicillium radicum* *, *Nigrospora oryzae* *, *Cladosporium* sp. *, *Nodulisporium* sp. * | [52] |
| *O. sativa* L. | Leaves | *Colletotrichum* spp. | [53] |
| *O. sativa* L. | Leaves | *Trichoderma* sp. | [53] |
| *O. sativa* L. | Leaves | *Penicillium* sp. | [53] |
| *O. sativa* L. | Leaves | *Chaetomium cupreum* | [53] |
| *O. sativa* L. | Leaves | *C. lunata* | [53] |
| *O. sativa* L. | Stems | *Aspergillus flavus* | [53] |
| *O. sativa* L. | Roots | *Rhizopus* spp. | [53] |
| *O. sativa* L. | Roots | *F. oxysporum* | [53] |
| *O. sativa* L. | Leaves | *Fusarium solani* | [53] |
| *O. sativa* L. | Leaves | *Colletotrichum* spp. | [53] |
| *O. sativa* L. | Roots | *A. flavus* | [53] |
| *O. sativa* L. | Roots | *Aspergillus niger* | [53] |
| *O. sativa* L. | Roots | *Pythium* spp. | [53] |
| *O. sativa* L. | Stems | *Trichoderma harzianum* | [53] |
| *O. sativa* L. | Stems | *Penicillium* sp. | [53] |
| *O. sativa* L. | Stems | *Chaetomium globosum* | [53] |
| *O. sativa* L. | Stems | *Chaetomium brasiliense* | [53] |
| *O. sativa* L. | Roots | *Penicillium simplicissimum* &,£,¢,¡ | [54] |
| *O. sativa* L. | Roots | *Trichoderma* sp. | [54] |
| *O. sativa* L. | Roots, stems | *Fusarium oxysporum* | [54] |
| *O. sativa* L. | Roots | *Aspergillus* sp. | [54] |
| *O. sativa* L. | Stems | *Acremonium* sp. ¢,£,¡ | [54] |
| *O. sativa* L. | Stems | *Phoma* sp. | [54] |
| *O. sativa* L. | Leaves | *Galactomyces geotrichum* | [54] |
| *O. sativa* L. | Leaves | *Penicillium* sp. | [54] |
| *O. sativa* L. | Leaves | *Aspergillus ustus* ¢,£,¡ | [54] |
| Wild rice (*O. rufipogon* Griff.) | Leaves | *C. Globosum* DX-THS3 # | [38,55] |
| *O. sativa* L. | Sheaths | *Nigrospora* sp. * | [57] |
| *O. sativa* L. | Sheaths | *Acremonium* sp. | [57] |

**Table 1.** *Cont.*

| Rice Variety | Phytonic Habitat | Endophytic Fungal Taxa | Reference |
|---|---|---|---|
| *O. sativa* L. | Sheaths | *Alternaria padwickii* | [57] |
| *O. sativa* L. | Sheaths | *Cephalosporium* sp. | [57] |
| *O. sativa* L. | Sheaths | *Chaetomium* sp. | [57] |
| *O. sativa* L. | Sheaths | *Curvularia lunata* | [57] |
| *O. sativa* L. | Sheaths | *Fusarium semitectum* | [57] |
| *O. sativa* L. | Sheaths | *Penicillium* sp. | [57] |
| *O. sativa* L. | Sheaths | *Pestalotia* sp. | [57] |
| *O. sativa* L. | Sheaths | *Phyllosticta* sp. | [57] |
| Upland rice | Roots | *Talaromyces* spp. [¢], *Penicillium* sp. [¢], *Trichocomaceae* sp. [¢], *Hypocreales* sp. [¢] | [60] |
| *O. sativa* L. | Roots | *Gaeumannomyces graminis* [&] | [79] |
| *O. sativa* L. | Roots | *Meyerozyma guilliermondii* [&] | [79] |
| *O. sativa* L. | Roots | *Gaeumannomyces amomi* [&] | [79] |
| *O. sativa* L. | Roots | *Phialemonium dimorphosporum* [&] | [79] |

Note: [*] Antagonistic activity. [&] Indoleacetic acid production and promotion of growth activity. [£] Siderophore production. [¢] P solubilization activity. [i] Alleviation of abiotic stress (salt tolerance, alleviation of heavy metal). [#] With a high potential to transform glycyrrhizin (GL) into glycyrrhetinic acid monoglucuronide (GAMG) (GL GAMG). [##] GL GAMG and glycyrrhetinic acid.

*5.2. Indirect Effects*

Plant-associated micro-organisms, which can affect host plant responses to environmental stress beneficially and enhance plant fitness under stress, benefit their hosts by providing nutrients, ameliorating biotic and abiotic stresses [95]. Endophytic fungi can play a crucial role in the sustainability of the host plant, enhancing the plant's tolerance of biotic and abiotic stresses, thereby benefiting plant growth [104–106].

5.2.1. Alleviation of Biotic Stress

The role of rice endophytic fungi in alleviating biotic stress mainly embodies the biological control of pathogenic micro-organisms and other pests. Fungal endophytes, as biocontrol agents against plant pathogens, widely exist in plants, which can help the host's health and disease resistance [14,52,107]. The mechanism of endophytic fungi in biological disease control includes direct inhibition of pathogenic bacteria activity through competition, antibacterial activity and fungal parasitism, and indirect inhibition of pathogenic bacteria activity through induced resistance and niche occupation [14,108,109].

A variety of endophytic fungi, such as *Paecilomyces tenuis*, *Talaromyces pinophilus*, *Nigrospora sphaerica*, *Nigrospora oryzae*, *Trichoderma longibrachiatum*, *Chaetomium*, *Fusarium*, and *Aspergillus terreus*, have been isolated from healthy rice tissues. They have antagonistic activity and can inhibit the incidence rate of rice brown spot disease (*Bipolaris oryzae*) and stimulate plant growth [35,39] (Table 1). After pre-inoculation with the endophytic fungus *P. Liquidambari*, the rice bakanae disease caused by *Fusarium proliferatum* was significantly decreased by inducing the defense responses of rice seedlings [110]. Endophytic fungi, such as *Antennariella placitae*, *Cladosporium* sp., *Alternaria* sp., *Trichoderma* sp., *Sarocladium* sp., *Dendryphiella* sp., *Phoma* sp., *Bipolaris* sp., *Penicillium* sp., *Fusarium* sp., and *Aspergillus* sp., exhibited significant antagonistic activity against multiple plant pathogenic fungi [38,44,111]. Endophytic fungi *Absidia* and *Acremonium* resulted in a particularly high inhibition of the growth of mycelia in rice blast pathogens [37]. Fungal endophytes of gramineous plants can reduce the preference of herbivores for insects, reduce herbivores' feeding on the host, and protect the host plants. *Melaspora* and *Beauveria* sp. isolated from rice leaves, sheaths, and stems enhanced the resistance of rice to brown plant hopper [57,112,113]. The endophytic fungus *Chaetomium brasiliense* isolated from rice stems in Thailand antagonizes Gram-positive bacteria by producing depsidones [114]. The endophyte *Fusarium moniliforme* strain Fe14 used for treating rice plants has shown potential antagonistic capacity against Meloidogyne graminicola through rice root exudates to repel nematode movement [115].

Also, the endophyte *Fusarium* sp. showed significant inhibition of the growth of *Rhizoctonia solani* when using the dual culture method [116]. Other studies showed that the endophytic yeast strains of Wickerhamomyces anomalus inhibited the growth of rice pathogenic fungi [117].

It cannot be ignored that the fungal volatile organic compounds (VOCs) produced by endophytic fungi are now being extensively applied to control plant pathogens. Many endophytic fungal strains, such as *Trichoderma* sp., *Macrophomina phaseolina*, *Fusarium* sp., *Setophoma terrestris*, *Paraphoma radicina*, *Oidiodendron cerealis*, *Plectosphaerella* sp., *Volutella* sp., *Cadophora* sp., *Phomopsis* sp., and *Colletotrichum* sp., can produce VOCs, which consist of compounds including simple hydrocarbons, heterocycles, aldehydes, ketones, alcohols, phenols, thioalcohols, thioesters and their derivatives, benzene derivatives, and cyclohexanes [107,118–123]. Due to their small size (<300 Da), high vapor pressure ($\geq$0.01 kPa at 20 °C), and hydrophobicity, VOC molecules can diffuse easily through plant cell membranes [124]. Therefore, they can play a significant role in the communication between fungi and their host in the ecosystem. Eleven endophytic fungi, represented by *Wickerhamyces anomalus* (seven species) and *Kodamae ohmeri* (four species), were screened from 407 endophytic yeast isolates from the tissues of rice, corn, and sugarcane and inhibited the growth of rice pathogenic fungi, such as *Curvularia lunata*, *Fusarium moniliforme*, and *Rhizoctonia solani*, by producing VOCs of mainly 3-methyl-1-butyl acetate and 3-methyl-1-butanol [117]. An interesting research study indicated that the VOCs of *Trichoderma longibrachiatum* EF5 isolated from the leaves of rice not only exert effects against fungal pathogens *Sclerotium rolfsii* and *Macrophomina phaseolina* but also exhibit growth-promoting properties in plants [125].

### 5.2.2. Alleviation of Abiotic Stress

The multiple attributes of endophytic fungi can play a pivotal role in improving rice growth under abiotic stresses [126] (Table 1). Endophytic fungi colonization in rice plants can improve the stress resistance of rice plants, such as salt resistance and drought resistance, reduce water consumption, and enhance plant growth rate and biomass [60,107–114,117,126–129]. After colonization of the salt-sensitive rice variety IR-64, the salt-tolerant endophytic fungus *Fusarium* sp. stimulated the growth of the host under salt stress and improved the salt tolerance of the host. Comparative transcriptome analysis shows that genes in rice plants colonized by *Fusarium* sp. are regulated, which are involved in both abiotic and biotic stress tolerance, and encode genes involved in signal perception and transduction [31]. Rice seeds were sown after being coated with *A. terreus*; afterward, 21-day-old seedlings were exposed to Sodium chloride (NaCl) concentrations and exhibited positive effects on salt tolerance in rice [130]. Dark septate endophytes fungi and *Metarhizium anisopliae* (MetA1) showed the capacity to resist salinity and water stress caused by water deficiency and enhanced rice plant growth [131,132]. A novel endophytic fungal strain from the root interior of an *Oryza coarctata* plant has been named as *Aspergillus welwitschiae* Ocstreb1 (AwOcstreb1), which improved the salt resistance of rice plants, significantly lowering the level of hydrogen peroxide ($H_2O_2$), electrolyte leakage, and Na+/K+ ratio under saline conditions, with higher expression of the salt overly sensitive 1 (SOS1) gene in rice plants [101]. A study on the drought-mitigating effects of rice plants mediated by the beneficial fungus *P. indica* under different moisture stresses showed that the colonization of beneficial fungi also increased lateral root yield, increased root volume, enhanced water use efficiency, and improved plant phosphorus uptake under both severe and non-stress conditions [133].

It has also been reported that the endophytic fungus *P. formosus* LWL1 alleviated heat stress and protected rice plants from long-term heat stress after inoculation of rice plants [100]. Some endophytic fungi have an antioxidant capacity, which can improve the drought resistance of rice under drought and semi-arid stress [60]. These endophytic fungi can effectively alleviate the damage to rice by secreting a coumarin analog (Z)-*N*-(4-hydroxystyryl) formamide under drought stress, which regulates the content of nicoti-

namide adenine dinucleotide phosphate (NADPH) oxidase, antioxidant, and heat shock protein [134]. The symbiotic relationship between the endophytic fungus *P. Liquidambari* and rice improved the biodegradation and remediation potential of phenanthrene in rice and promoted food safety [135,136].

Cadmium (Cd) and arsenic (AsV) are the factors that pose a threat to agriculture and human health [137,138]. Therefore, reducing grain Cd and As in rice is crucial for over half of the world's population. The experimental results indicated that the plant-growth-promoting endophytic fungi (PGPEF) had a significant promoting effect on the growth of rice seedlings under cadmium stress [72]. An endophytic fungus *Falciphora oryzae* isolated from *Oryza granulata* (wild rice) was shown to decrease cadmium (Cd) accumulation in the roots [18]. Endophytic *Fusarium* Fo10 exhibits tolerance to high levels of Cd, and its symbiosis in rice is regulated by OsCAL1 (Cd accumulation in leaf 1) dynamics, coordinating the plant's uptake of cadmium. This process relies on the salicylic acid signaling pathway to maintain low cadmium levels in the cytosol of the rice host, thereby reducing cadmium levels in the grains [139]. Endophytic fungus *Serendipita indica* colonization in the rice root can dramatically enhance the detoxification of arsenic in rice plants, recovering chlorophyll content and growth, improving photosynthesis, transpiration, and water use efficiency under arsenic stress [140–143].

Endophytic fungus *Piriformospora indica* colonization can protect the photosynthetic machinery of the rice plant from arsenic stress by adjusting the antioxidative enzyme system, reducing the availability of free arsenic in the rice plant, improving the growth performance of rice under Fe-limited conditions, and conferring salinity stress tolerance to the rice plant [144,145].

## 6. Secondary Metabolites of Endophytic Fungi

Endophytic fungi are considered prolific sources of bioactive compounds or sources, which can be used for the synthesis of bioactive natural products [38,47,146]. Filamentous fungi can produce a variety of secondary metabolites with bioactive compounds, mainly including polyketides, non-ribosomal peptides, alkaloids, and terpenes. These metabolites are considered rich sources of biomolecules with potential medicinal value [147–149]. Fungal endophytes may produce some bioactive metabolites with special structural characteristics because of their special living environment. In particular, endophytic fungi have the same ability to produce compounds as medicinal plants and exhibit antioxidant and antibacterial properties. They have additional functions that include the secretion of secondary metabolites, such as alcohol and xylitol, to help their hosts alleviate biotic stress [150]. Currently, endophytic fungi are viewed as novel biocatalyst resources and have become the focus of new drugs, biological pesticides, and biological control agents [11,47,146,151–155]. Some strains of endophytic fungi isolated from rice plants have been shown to have the potential to transform glycyrrhizin into glycyrrhetinic acid monoglucuronide or glycyrrhetinic acid [50] (Table 1). They produce different new secondary metabolites, especially volatile organic compounds, which can protect the host from pathogens and pests and improve the post-harvest storage of crops [16]. Other endophytic fungi isolated from rice can produce cellulose, chitin, lignin, pectinase, soluble metabolites, aromatic nitro compounds, and volatile metabolites and exhibit antagonistic activities, which have significant inhibitory effects on rice plant pathogens [76,156].

## 7. Conclusions and Perspectives

A large diversity of endophytic fungi in rice plants interact with the host to maintain mutual benefit, which is vital in the functioning of ecosystems. Statistical analysis based on the Origin software 2021 insightfully showed that five phyla and more than sixty genera of endophytic fungal communities have been identified from rice parts through culture-dependent methods. *Ascomycota* is the main endophytic fungal community in rice. Molecular techniques, such as high-throughput amplicon sequencing (HTAS), and bioinformatics tools are necessary and reliable in identifying fungal endophytes and exploring

the community structure of endophytic fungi in rice plants, especially unculturable fungi. Generally, the combination of culture-dependent and culture-independent methods will provide more powerful insights into the complete information on microbial communities. The diversity and community abundance of fungal endophytes are affected by many factors, such as the genotype of the host plant, the conditions of the soil, the period of rice plant's growth and development, and the interaction between micro-organisms and the host plant. Specifically, rice plants provide a living environment and nutrition for fungal endophytes. Accordingly, the feedback between fungal endophytes and the host caused by stimulation of the production of metabolites effectively promotes the growth and development of rice plants and responds to external biotic and abiotic stresses through metabolic mechanisms, playing a crucial role in ecological agriculture. Endophytic fungi in rice have multiple functions to enhance the host's absorption of mineral nutrients, secrete plant hormones, and help the host resist diseases, pests, salt, drought, and heavy metal stress through direct and indirect interactions. The plant–endophyte interaction system has demonstrated promising potential for pollution remediation in agriculture.

Currently, the knowledge of rice fungal endophytes is still limited, especially in terms of how they interact with the host plant, how signal transduction reacts to activating the defense response in the host plant, and which type of nutrient is provided by the host plant; the molecular mechanism needs to be explored for an enhanced understanding. The interaction between fungal endophytes and bacterial endophytes and the host plant deserves future investigations. In addition, fungal endophytes in rice plants as a new resource for biomedicine development may produce some new functional secondary metabolites, especially functional bioactive substances, which may supplement the current drugs used to treat human miscellaneous diseases, and it remains necessary to explore these in rice plants. More attention should be focused on the screening, identification, and production of functional fungal endophytes. However, most endophytes that are unculturable should not be disregarded, as the metagenome of endophytes is of great importance for research. Of course, new methods for the culturing of some novel endophytes warrant further investigations. From the perspective of the functional genes of fungal endophytes, the focus should be on the great potential of screening recombinant strains through bioengineering technology or the CRISPR/Cas9 system to create strong functional medicinal products with anticancer, antioxidant, and antibacterial properties.

**Author Contributions:** B.L. and Z.L. provided guidance and designed the review manuscript; Y.H. wrote the review paper; D.L. and H.L. helped to improve the manuscript language; G.L. and M.H. helped to revise the manuscript. All authors have read and agreed to the published version of the manuscript.

**Funding:** This work was supported by the National Engineering Research Center of JUNCAO Technology, Fujian Agriculture and Forestry University (No. 271-KH200039A).

**Acknowledgments:** We are grateful to Peng-hu Liu for his critical comments and Jun-xin Yao for his help in improving the photographs.

**Conflicts of Interest:** All authors declare no conflicts of interest.

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
