# Peer review of "Endophytic Fungi in Rice Plants and Their Prospective Uses"

_2036-7481, doi:10.3390/microbiolres15020064_

Round 1

Reviewer 1 Report

Comments and Suggestions for Authors

The manuscript titled "Endophytic Fungi in Rice Plants and Their Prospective Uses" is obviously an important and useful review for future researchers studying endophytic fungi in rice and other grasses.

Here I make some suggestions to enrich the review.

1. Include in the introduction subject about the Fusarium fujikuroi complex associated with grasses such as wild rice in Australia and native grasses in Brazil (Brachiaria)

2. Include the importance of the species of this Fusarium complex (Many species are producers of phytominines)

3. Include a paragraph on volatile substances produced by fungi, including many of these substances that are potential for use in controlling phytopathogenic fungi

4. Paragraph on fungus isolation in rice, I found it extremely long... summarize

Furthermore, I see that it is a good review and deserves to be published, after the suggested revisions.

Comments on the Quality of English Language

Yes

Reviewer 2 Report

Comments and Suggestions for Authors

The “Endophytic Fungi in Rice Plants and Their Prospective Uses” review paper was interesting but needed some improvement in writing. The author explained the theory not in full detail; perhaps the whole article is like the introduction part. Results or statistical variances were not provided; hence, mentioning the reference in between doesn’t complete the sentence.

The minor corrections are given below.

The space consistency between text and reference number should follow the same pattern. If any software is used, please mention the version.

Page 2, line no. 5 Change from “lebsiella” to “Lebsiella”.

Page 2 line 4 space between B-89 and comma, delete the space.

On page 2, line 15, give space between crops [8, 27]. Check the correct spacing throughout the article.

Page 2, last paragraphs: Give space between into and operational.

Page 2, last line, “To further identify….[34, 38] rephrase the sentence.

Page 3, Line 12, change the word gene “pollution.”

Page 3 line 15 change ednophytes to “endophytes”

page 3 Line 16 yeast. Not yeast,

Page 3 Wang W et al  is the author initial is needed? Pls check and change through out manuscript.

Page 3 last para endophtes to “endophytes”

Some to “Some”

Page 6 Ph. liquidambari  to P. liquidambari

Page 7 delete and inbetween “molybdenum and selenium”

“Rice plants inoculated with Absidia and Cylindrocladium endophytic fungal isolates.

showed significant increases in growth[71]”  what is the significance? The author must complete the result explanation, many area the author just give the reference no. which is harder to conclude the results and importance of the research work.

Page 7 5.2 section “The interaction….. “rewrite the sentence.

Figure 4 caption change endophte to endophyte.

Overall, the language must be checked and should be modified for better publication. Many places it was like translated version. Thoroughly check each sentence, space and line spacing.

Round 2

Reviewer 2 Report

Comments and Suggestions for Authors

The manuscript can be accepted for publication